# Modification of Tomato Photosystem II Photochemistry with Engineered Zinc Oxide Nanorods

**DOI:** 10.3390/plants12193502

**Published:** 2023-10-08

**Authors:** Panagiota Tryfon, Ilektra Sperdouli, Ioannis-Dimosthenis S. Adamakis, Stefanos Mourdikoudis, Catherine Dendrinou-Samara, Michael Moustakas

**Affiliations:** 1Laboratory of Inorganic Chemistry, Department of Chemistry, Aristotle University of Thessaloniki, 54124 Thessaloniki, Greece; tryfon.giota@gmail.com; 2Institute of Plant Breeding and Genetic Resources, Hellenic Agricultural Organization-Dimitra, 57001 Thessaloniki, Greece; ilektras@bio.auth.gr; 3Section of Botany, Department of Biology, National and Kapodistrian University of Athens, 15784 Athens, Greece; iadamaki@biol.uoa.gr; 4Biophysics Group, Department of Physics and Astronomy, University College London, London WC1E 6BT, UK; s.mourdikoudis@ucl.ac.uk; 5UCL Healthcare Biomagnetics and Nanomaterials Laboratories, 21 Albemarle Street, London W1S 4BS, UK; 6Separation and Conversion Technology, Flemish Institute for Technological Research (VITO), Boeretang 200, 2400 Mol, Belgium; 7Department of Botany, Aristotle University of Thessaloniki, 54124 Thessaloniki, Greece

**Keywords:** chlorophyll fluorescence, maximum PSII photochemistry (F*v*/F*m*), nanoparticles, reactive oxygen species, photoinhibition, oxygen-evolving complex, nanoagrochemicals, electron transport rate, effective quantum yield of PSII photochemistry (Φ*_PSII_*)

## Abstract

We recently proposed the use of engineered irregularly shaped zinc oxide nanoparticles (ZnO NPs) coated with oleylamine (OAm), as photosynthetic biostimulants, to enhance crop yield. In the current research, we tested newly engineered rod-shaped ZnO nanorods (NRs) coated with oleylamine (ZnO@OAm NRs) regarding their in vivo behavior related to photosynthetic function and reactive oxygen species (ROS) generation in tomato (*Lycopersicon esculentum* Mill.) plants. ZnO@OAm NRs were produced via solvothermal synthesis. Their physicochemical assessment revealed a crystallite size of 15 nm, an organic coating of 8.7% *w*/*w*, a hydrodynamic diameter of 122 nm, and a ζ-potential of −4.8 mV. The chlorophyll content of tomato leaflets after a foliar spray with 15 mg L^−1^ ZnO@OAm NRs presented a hormetic response, with an increased content 30 min after the spray, which dropped to control levels 90 min after the spray. Simultaneously, 90 min after the spray, the efficiency of the oxygen-evolving complex (OEC) decreased significantly (*p* < 0.05) compared to control values, with a concomitant increase in ROS generation, a decrease in the maximum efficiency of PSII photochemistry (F*v*/F*m*), a decrease in the electron transport rate (ETR), and a decrease in the effective quantum yield of PSII photochemistry (Φ*_PSII_*), indicating reduced PSII efficiency. The decreased ETR and Φ*_PSII_* were due to the reduced efficiency of PSII reaction centers (F*v’*/F*m’*). There were no alterations in the excess excitation energy at PSII or the fraction of open PSII reaction centers (q*p*). We discovered that rod-shaped ZnO@OAm NRs reduced PSII photochemistry, in contrast to irregularly shaped ZnO@OAm NPs, which enhanced PSII efficiency. Thus, the shape and organic coating of the nanoparticles play a critical role in the mechanism of their action and their impact on crop yield when they are used in agriculture.

## 1. Introduction

Nanoparticles (NPs) have been shown to have great potential to improve the agri-food sector and revolutionize agriculture [1]. They can act as fertilizers, improving plant performance and protecting crop plants from environmental stresses. They can also suppress plant disease and increase crop yield [1,2,3]. NPs can be used to diminish the amounts of sprayed chemical products, reduce nutrient losses in fertilization, and drive precision and sustainability in agriculture [1,4]. However, some metal-based NPs have been reported to decrease plant growth and performance by causing phytotoxic effects in crops [5,6]. NPs cause oxidative stress that depends on the concentration, reactivity, surface area, and size used as well as the plant age and the treated plant species [7,8].

Zinc oxide (ZnO) is recognized for its biocompatibility, biodegradability, and safety in both medical and environmental contexts [9]. Engineered zinc oxide nanoparticles (ZnO NPs) have been successfully applied as novel nano-fertilizers for crop improvement and to enrich Zn-deficit soil with Zn [10]. The application of ZnO NPs can enhance crop yield, providing an effective solution for nutrient supplementation in farming systems, and these NPs were characterized as a novel alternative for sustainable agricultural development [11,12]. It has been proposed that ZnO NPs can be used as photosynthetic biostimulants for enhancing crop yields [13], as they possess insecticidal [14], antifungal [15], and antibacterial properties [16]. The impact of ZnO NPs on metabolic attributes and the photosynthetic function of rice seedlings in hydroponic culture was found to be concentration-dependent when improving the qualitative and quantitative characteristics of the plants [17]. Foliar sprays with 20 and 100 mg L^−1^ ZnO NPs enhanced tomato growth by increasing the chlorophyll content and photosystem II (PSII) activity [18]. Similarly, ZnO NPs were found to amplify the enzymatic functions in *Lycopersicon esculentum*, triggering the defense mechanisms of the plant [19,20]. ZnO NPs have a favorable effect on plant growth and development in ordinary amounts [21]. Different concentrations of ZnO NPs showed various effects on the growth and development of *Arabidopsis thaliana* [22,23]. The response of *Stevia rebaudiana* to ZnO NPs also displayed a dual nature, exhibiting both favorable and adverse effects depending on the NP concentration [24]. Exposing *Hordeum vulgare* seedlings to ZnO NPs for 7 days reduced root growth in a dose-dependent manner [25]. A damaging impact is induced by replacing ions of other elements, causing the generation of ROS [21]. A negative impact of ZnO NPs through ethylene has been recorded in rice seedlings, causing ultrastructural and stomatal damage [26]. However, ZnO NPs have been reported to decrease the growth of spring barley through the modulation of oxidative stress and antioxidant enzyme metabolism [21].

A positive impact of ZnO NPs on drought- and salt-stressed plants was recorded. ZnO NPs at 25 and 50 mg L^−1^ concentrations were proven to be the optimal treatments for alleviating drought-induced oxidative stress in tomato plants [27]. In salinity-stressed bean plants, a foliar spray of biosynthesized 50 mg L^−1^ ZnO NPs helped plants to overcome the negative effects of salinity by increasing enzymatic and non-enzymatic antioxidants that reduced oxidative stress and increased soluble sugars, glycine betaine, proline, and amino acids [8]. A foliar spray of 200 mg L^−1^ ZnO NPs decreased the adverse effects of salinity on both photosystems (PSI and PSII) of *Pisum sativum* as well as on stomatal closure, pigment content, and membrane integrity [28]. However, ZnO NPs could improve the chlorophyll content, wheat growth, and wheat grain yield under salt stress [29] and mitigate salinity stress in rice seedlings by upregulating the antioxidative system and enhancing physiological and biochemical indices [30]. Significant improvements in wheat and rice growth and grain yield under salinity stress using ZnO NPs were also recorded by Mazhar et al. [31]. A foliar application of 16–35 nm ZnO NPs at 10 mg L^−1^ increased the activity of superoxide dismutase (SOD) and catalase (CAT) and the concentration of photosynthetic pigments in *Abelmoschus esculentus* L., while it reduced the total soluble sugars and proline [32].

The early-stage recognition of any stress with imaging techniques permits the avoidance of plant injury and consequently important crop losses [33,34,35]. Chlorophyll-*a* fluorescence imaging is among the best-qualified imaging techniques to estimate plant health. It also allows the pre-symptomatic monitoring of plant stress symptoms [36,37,38]. Chlorophyll-*a* fluorescence analysis is a capable tool for evaluating NP effects on plant function and to monitor plant−NP relations [2,13,39].

We recently proposed that engineered irregularly shaped ZnO NPs coated with oleylamine (ZnO@OAm NPs) can be effectively used to boost photosynthetic function and thus are appropriate for enhancing crop yield [13]. Nevertheless, the presented literature suggests that plant responses to zinc-based NPs are multifactorial and are influenced by factors such as the plant species, NP morphology and size, the applied dosage, and the application method. Thus, the purpose of the current research was to compare the newly engineered rod-shaped ZnO@OAm NRs with the previously examined irregularly shaped ZnO@OAm NPs for their behavior related to photosynthetic function and ROS generation. Since the impact of ZnO NPs on plants is influenced by their morphology and size, we hypothesized that the rod-shaped ZnO@OAm NRs would have a different mechanism of action than irregularly shaped ZnO@OAm NPs and therefore a different impact on crop yield; thus we decided to investigate it in the current study, as demonstrated below.

## 2. Results

### 2.1. Synthesis and Characterization of Zinc Oxide Nanorods

The crystal structure of the ZnO@OAm NRs was identified via XRD analysis. The XRD diffractogram displayed the primary diffraction peaks at 31.6° (100), 34° (002), and 36° (101) (Figure 1a), which were assigned to the wurtzite structure of zinc oxide (JCPDS card #89-0510) [40]. The crystallite size was calculated to be 15 nm. The percentage and profile of the organic coating on the ZnO@OAm NRs were determined through TGA, and the results illustrated the gradual decomposition of OAm (Figure 1b). In the initial stage, absorbed water molecules were removed (0.84% *w*/*w*) up to 150 °C, followed by a weight loss of 7.7% *w*/*w* in the subsequent stage due to OAm. The final stage evidenced a mass loss of 1% *w*/*w*, also linked to the removal of OAm from the metallic cores of the particles. The total weight loss attributed to OAm amounted to 8.7% *w*/*w*.

An analysis of the TEM images revealed that the particles possessed a distinctive rod-like structure (Figure 2a). To quantify the dimensions of these particles, we conducted a histogram analysis. The results illustrated that the ZnO@OAm NRs had an average width of 22 ± 2.1 nm (Figure 2b). Furthermore, the length of the NRs was measured, yielding an average value of 62 ± 1.3 nm.

Optical characterization of the ZnO@OAm NRs, when dispersed in an ethanol/water mixture (1:3 ratio), was conducted using UV-Vis spectroscopy. The UV-Vis absorbance spectrum (Appendix A) prominently showcased an absorbance peak situated at 367 nm. Employing the Tauc plot, the band gap of the NRs was ascertained to be 3.16 eV.

ZnO@OAm NRs exhibiting hydrophobic characteristics were dispersed in an ethanol/water mixture (1:3 ratio) for DLS analysis. The hydrodynamic diameter, as discerned, was 122 ± 0.42 nm (Appendix A), signifying monodispersity, with a PDI value of 0.17, which underscored the colloidal stability of the system. Furthermore, the ζ-potential measurements provided a surface charge for the nanoparticles at −4.8 ± 0.31 mV (Appendix A).

### 2.2. Impact of Zinc Oxide Nanorods on Chlorophyll Content of Tomato Leaflets

The chlorophyll content increased significantly (*p* < 0.05) 30 min after spraying with ZnO@OAm NRs, but 90 min after the spray it returned to control values (Figure 3). Thus, the response of the chlorophyll content to ZnO@OAm NRs resembled a hormetic response.

### 2.3. Changes in PSII Photochemistry due to Zinc Oxide Nanorods

#### 2.3.1. Efficiency of the Oxygen-Evolving Complex and Maximum Efficiency of PSII Photochemistry

The efficiency of the oxygen-evolving complex (OEC) did not change 30 min after spraying with ZnO@OAm NRs, but 90 min after the spray it decreased significantly (*p* < 0.05) compared to the corresponding control values (Figure 4a). A similar response pattern was noticed in the maximum efficiency of PSII photochemistry (F*v*/F*m*) (Figure 4b).

#### 2.3.2. Allocation of Absorbed Light Energy and Fraction of Open PSII Reaction Centers

Absorbed light energy is distributed to PSII photochemistry (Φ*_PSII_*), regulated non-photochemical energy loss in PSII (Φ*_NPQ_*), and non-regulated energy loss in PSII (Φ*_NO_*), with Φ*_PSII_* + Φ*_NPQ_* + Φ*_NO_* equal to 1 [41]. The effective quantum yield of PSII photochemistry (Φ*_PSII_*) did not change 30 min after spraying with ZnO@OAm NRs, but 90 min after the spray, it decreased significantly (*p* < 0.05) compared to the corresponding control values under both the growth-light (GL, 600 μmol photons m^−2^ s^−1^) and high-light (HL, 1000 μmol photons m^−2^ s^−1^) actinic light (AL) intensity conditions (Figure 5a). Under the GL intensity, all Φ*_PSII_* values were significantly (*p* < 0.05) higher compared to the corresponding HL intensity values (Figure 5a).

The regulated non-photochemical energy loss as heat in PSII (Φ*_NPQ_*) did not change 30 or 90 min after spraying with ZnO@OAm NRs under both the GL and HL intensity conditions (Figure 5b). Under HL intensity, all Φ*_NPQ_* values were significantly (*p* < 0.05) higher compared to the corresponding GL intensity values (Figure 5b). The non-regulated energy loss in PSII (Φ*_NO_*) at the GL intensity increased significantly (*p* < 0.05) both 30 and 90 min after spraying with ZnO@OAm NRs, but at the HL intensity it only increased significantly (*p* < 0.05) 90 min after spraying with ZnO@OAm NRs (Figure 5c).

The redox state of quinone A (Q_A_), or in other words the fraction of open PSII reaction centers, which indicates photochemical quenching (q*p*), did not change 30 or 90 min after spraying with ZnO@OAm NRs under both the GL and HL intensity conditions (Figure 5d).

#### 2.3.3. Non-Photochemical Quenching and Electron Transport Rate

The parameter of non-photochemical quenching (NPQ), reflecting the dissipation of excess excitation energy as heat, did not change 30 or 90 min after spraying with ZnO@OAm NRs under both the GL and HL intensity conditions (Figure 6a). The electron transport rate (ETR) did not change 30 min after spraying with ZnO@OAm NRs, but 90 min after the spray, it decreased significantly (*p* < 0.05) compared to the corresponding control values under both the GL and HL intensity conditions (Figure 6b).

#### 2.3.4. Efficiency of PSII Reaction Centers and Excess Excitation Energy

The efficiency of PSII reaction centers remained the same as in the controls 30 min after spraying with ZnO@OAm NRs, but 90 min after the spray, it decreased significantly (*p* < 0.05) compared to the corresponding control values under both the GL and HL intensity conditions (Figure 6c).

The excess excitation energy at PSII did not differ from controls 30 or 90 min after spraying with ZnO@OAm NRs under both the GL and HL intensity conditions (Figure 6d).

### 2.4. Impact of Zinc Oxide Nanoparticles on H_2_O_2_ Generation

90 minutes after spraying with 15 mg L^−1^ ZnO@OAm NRs, hydrogen peroxide (H_2_O_2_) formation was observed on leaf midribs and minor tomato lamina veins (Figure 7b). At the same time, in water-sprayed tomato leaflets, no H_2_O_2_ could be observed (Figure 7a).

## 3. Discussion

ZnO NPs can be prepared in a very wide variety of sizes and shapes. Common ZnO NP morphologies like rods, spheres, flowers, and stars exhibit unique physicochemical properties influenced by their shapes and sizes [42]. Various synthetic approaches, including solvothermal and hydrothermal ones, produce ZnO NPs of distinct sizes and shapes [43]. Notably, rods and wires penetrate bacterial walls more effectively compared to spherical forms, while flower-shaped ZnO NPs show superior biocidal activity against *E. coli* and *S. aureus* [44,45]. Recent works, such as the study by Wang et al. [46], have explored the interplay of ZnO NPs with plant photosynthesis, highlighting their promise for agricultural and environmental applications. In the present investigation, ZnO NRs that were coated with OAm were effectively produced using the polyol-solvothermal synthesis approach, which has gained prominence due to its simplicity and efficiency [47].

Each polyol exhibits reductive capabilities that vary based on its molecular weight, leading to the creation of diverse intermediate complexes with the initial materials. This mechanism is influenced by selected reaction conditions, such as temperature, time, and pressure, as well as the inclusion of surfactants. Therefore, it results in NPs with varying sizes and shapes [48,49,50]. Thus, in the present work, by using the lower-molecular-weight diethylene glycol in the presence of OAm and an 8 h reaction time, well-formed rods of ZnO@OAm NRs of 15 nm crystallite size were isolated (Figure 1a). Earlier findings [51] showed that in the case of TrEG and an 8 h reaction time, marginally bigger crystallites of 18 nm, 38 nm width, and 83 nm length were formed, while in the sole presence of OAm irregularly shaped NPs of 19 nm were isolated. It has been noted that NPs with diameters below 50 nm, akin to our ZnO@OAm NRs, generally follow the plastid transport route within plants. In contrast, particles sized between 50 and 200 nm predominantly utilize the apoplast pathway [52].

Additionally, based on the TGA results, the ZnO@OAm NRs (Figure 1b) were minimally coated with OAm (8.7% *w*/*w*), being hydrophobic. Nonpolar nanomaterials mainly exhibit hydrophobic characteristics due to the dominance of *van der Waals* forces [53]. Notably, the hydrophobicity of NPs significantly impacts their interactions within biological and environmental systems, influencing factors like protein and lipid corona formation [54], immune responses [55], cellular uptake [56], bioaccumulation, and bioavailability [57].

Considering the optical properties, the primary absorption peak (Appendix A) exhibited a noticeable blue shift that was predominantly ascribed to the Burstein–Moss effect [58,59]. This phenomenon is frequently seen in n-type semiconductors [60] and is associated with reduced size, quantum confinement effects [61,62], impediments in low-energy transitions [63], and alterations in surface morphology [64,65].

Rod-shaped nanostructures have certainly a higher aspect ratio compared to spherical and/or irregular NPs [66,67]. This difference in aspect ratio can influence their properties, including the way that they interact with cells and other biological systems [68]. For instance, rod-shaped gold NPs, despite being analogous in size to their spherical counterparts, exhibited enhanced absorption and integration [69]. In contrast, these rod-shaped particles exhibited reduced exocytosis compared to their spherical equivalents [70]. Elongated particles, due to their increased aspect ratios, demonstrate a more pronounced cellular uptake compared to spherical NPs of similar size [71,72,73]. The distinct morphologies of NPs, coupled with their unique interfacial properties, lead to nuanced surface interactions that ultimately dictate their absorption kinetics [74,75].

ZnO NPs can provoke ROS generation, and thus they are considered to exhibit anticancer and antibacterial activities [76,77,78]. In our work, the synthesized ZnO@OAm NRs decreased the efficiency of the OEC (Figure 4a) and increased ROS production (Figure 7b). Enhanced ROS production inhibits the repair of photodamaged PSII and especially the de novo synthesis of the D1 protein [79,80], thus reducing the efficiency of PSII reaction centers (F*v’*/F*m’*) (Figure 6c). A decreased Φ*_PSII_* can be assigned either to a decreased fraction of open PSII reaction centers (q*p*) or a decreased efficiency of reaction centers (F*v*′/F*m*′) [81]. In our case, the decreased Φ*_PSII_* was attributed to the reduced efficiency of the reaction centers (F*v*′/F*m*′) (Figure 6c) since the fraction of open PSII reaction centers (q*p*) remained unaltered (Figure 5d). Thus, the reduced efficiency of PSII reaction centers (F*v*′/F*m*′) resulted in a reduced electron transport rate (ETR) (Figure 6b) and a decreased effective quantum yield of PSII photochemistry (Φ*_PSII_*) (Figure 5a).

The decrease in the effective quantum yield of PSII photochemistry (Φ*_PSII_*) 90 min after spraying with ZnO@OAm NRs (Figure 5a) was accompanied by a significant (*p* < 0.05) increase in the non-regulated energy loss in PSII (Φ*_NO_*) at both GL and HL intensities (Figure 5c). The increase in Φ*_NO_*, which is a measure of the singlet oxygen (^1^O_2_) generation [82,83,84], suggests an increase in ROS production. When the singlet excited chlorophyll state (^1^Chl*) is not quenched, the triplet chlorophyll state (^3^Chl*) is developed through intersystem crossing, which leads to singlet oxygen (^1^O_2_) formation [85,86,87,88]. ROS generation is prevented by downregulating ^1^Chl* through the mechanism of non-photochemical quenching (NPQ), quenching ^3^Chl*, or the antioxidant mechanisms that scavenge ROS and improve fitness [89,90,91,92]. However, in tomato leaves sprayed with ZnO@OAm NRs, the photoprotective mechanism of NPQ did not increase (Figure 6a) to prevent ROS production [89,90,93,94]. As a result, ^1^O_2_ (Figure 5c) and H_2_O_2_ (Figure 7b) were created, suggesting increased ROS generation.

ZnO@OAm NRs resulted in reduced efficiency in the OEC (Figure 4a), which caused photoinhibition, as judged by the decrease in the maximum photochemistry (F*v*/F*m*) (Figure 4b). A decline in OEC activity leads to a rapid decline in F*v*/F*m*, and the donor-side-induced photoinhibition further reduces F*v*/F*m* [95,96]. The production of H_2_O_2_ on leaf midribs and minor tomato lamina veins that were sprayed with ZnO@OAm NRs (Figure 7b) was probably due to donor-side photoinhibition as the result of the OEC malfunction [97]. The H_2_O_2_ production observed in the current research was in agreement with the increased ROS generation on the leaf midribs and minor lamina veins of *Brassica juncea* treated with ZnO NPs [98]. It has to be noted that the generation of H_2_O_2_ predominantly depends on the ZnO NP surface, leading to the production of more ROS [99].

Upon exposure to light, ZnO NPs release oxygen molecules from their active sites, leading to the generation of various ROS on the surface of a ZnO nanocrystal [99]. ROS accumulation, due to the absence of the photoprotective mechanism of NPQ, is the result of the donor-side photoinhibition that is associated with H_2_O_2_ formation, which can be oxidized to the superoxide radical (O_2_^•−^) or reduced to the hydroxyl radical (HO^•^) [97]. When oxygen molecules bind to the rod-shaped ZnO NPs’ surfaces, they trap free electrons, as described by the equation O_2_(*g*) + e^−^ → O_2_^−^(*ad*) [100]. Once the UV light source is discontinued, the concentration of holes becomes outnumbered by electrons within the ZnO NRs. As these holes recombine with the electrons, the oxygen molecules once again adhere to the surface, capturing the free electrons, which results in the conductance dropping [100].

Despite their destructive action, ROS are described as second messengers in a plethora of cellular and developmental processes [88,101,102,103,104]. ROS have been shown to diffuse through leaf veins and act as long-distance signaling molecules [105,106,107,108,109,110]. In fact, ROS primarily act by inhibiting the repair of photodamaged PSII reaction centers and not by damaging PSII reaction centers directly [111].

ZnO NPs enter leaves through stomata. They are subsequently transported through the apoplast to mesophyll cells and are then translocated to chloroplasts [112,113,114]. However, after leaf entry the long-distance transport of foliar-applied ZnO NPs is not well understood [114]. In chloroplasts, ZnO NPs alter PSII activity, measured as the maximum efficiency of PSII photochemistry (F*v*/F*m*) [115]. In mesophyll cells, ZnO NPs have been reported to decrease δ-aminolevulinic acid (ALA), indicating changes in the chlorophyll metabolic pathway [21]. Foliar sprays with 20 and 100 mg L^−1^ ZnO NPs enhanced the chlorophyll content in tomatoes [18], while long-term foliar application of 3 mg L^−1^ ZnO NPs increased both chlorophyll a (Chl a), and chlorophyll b (Chl b) in tomatoes [116]. Faizan et al. [19] reported that ZnO NPs increased the chlorophyll contents of tomato plants in a dose- and exposure-time-dependent way. In wheat plants under salt stress, the application of ZnO NPs increased the amounts of Chl a and Chl b by 24.6% and 10%, respectively [29]. Our data show that the response of the chlorophyll content to ZnO@OAm NRs resembled a hormetic response, with an increased content for a short period that decreased afterwards to the control level (Figure 3). A disturbance of homeostasis results in a hormetic stimulation as an overcompensation reaction [117,118]. Lately, hormesis has been revealed to occur in several organisms independent of the stressor or the biological function being examined [84,119,120,121,122].

## 4. Materials and Methods

### 4.1. Synthesis of ZnO@OAm NRs

All the reagents used for the synthesis of ZnO@OAm NRs were of analytical grade. The synthesis of hydrophobic ZnO@OAm NRs was adapted from a previously reported polyol-solvothermal approach, with a modification in the glycol used. Specifically, DEG was employed as the solvent instead of the triethylene glycol (TrEG) used in an earlier study [51]. In the modified procedure, 1.06 mmol Zn(acac)_2_ was dissolved in a mixture of 4 mL of DEG and 4 mL of OAm. The solution was agitated at 30 °C for 15 min before being transferred to a Teflon-lined autoclave. The reaction was carried out under autogenous pressure at 200 °C for 8 h. Subsequently, the mixture was centrifuged at 5000 rpm for 20 min, followed by rinsing with ethyl alcohol four times to remove any untreated precursors.

### 4.2. Characterization of ZnO@OAm NRs

The crystal size and structure of the NRs were determined via X-ray diffraction (XRD) on a Philips PW 1820 diffractometer with monochromatized Cu Kα radiation (λ = 1.5406 Å) in the range of 2θ from 20 to 90°. The morphology and particle size were analyzed using a JEOL JEM 1200-EX (Tokyo, Japan) transmission electron microscope (TEM). The FT-IR spectra (ranging from 4000 to 450 cm^−1^) of the NRs were obtained using a Nicolet iS20 spectrometer equipped with a monolithic diamond ATR crystal (Thermo Fisher Scientific, Waltham, MA, USA). The thermal stability and the amount of the organic coating were evaluated via thermogravimetric analysis (TGA) using a SETA−RAM SetSys-1200 (KEP Technologies, Caluire, France) under a N_2_ atmosphere with a heating rate of 10 °C min^−1^. The optical properties of the NRs were investigated using a UV-Vis spectrophotometer (V-750, Jasco, Tokyo, Japan). The hydrodynamic size (nm), polydispersity (PDI), and ζ-potential (mV) of dispersed NRs in an ethanol/water solution (1:3 ratio) were determined via dynamic light scattering (DLS) at 25 °C using a Zetasizer (Nano ZS Malvern apparatus VASCO Flex™ Particle Size Analyzer NanoQ V2.5.4.0, Malvern, UK).

### 4.3. Plant Material and Growth Conditions

To test the impact of the synthesized ZnO@OAm NRs on photosynthetic function, we purchased tomato (*Lycopersicon esculentum* Mill. cv. Galli) plants from the market. The plants were left for three days to acclimate in a greenhouse at 25 ± 1/20 ± 1 °C day/night temperatures, with a 14 h day/night photoperiod provided by a photosynthetic photon flux density (PPFD) of 600 ± 10 μmol quanta m^−2^ s^−1^ and relative day/night humidities of 65 ± 5/75 ± 5%.

### 4.4. Exposure of Tomato Plants to ZnO@OAm NRs

After the three-day acclimation period, tomato plants were foliar-sprayed once with 15 mg L^−1^ ZnO@OAm NRs or double-distilled water (control). Each plant received 10 mL of 15 mg L^−1^ ZnO@OAm NRs or 10 mL of double-distilled water. Three to five plants were used in each treatment with three independent replicates.

### 4.5. Chlorophyll Content Measurements

The chlorophyll contents of the control plants and the ZnO@OAm NR-treated ones were estimated with a portable Chlorophyll Content Meter (Model Cl-01, Hansatech Instruments Ltd., Norfolk, UK) [123,124]. Six measurements were recorded around the middle leaf from each treatment, and the measurements were averaged from all biological replicates.

### 4.6. Chlorophyll Fluorescence Analysis

The impact of the synthesized ZnO@OAm NRs on PSII function was evaluated in dark-adapted (20 min) tomato leaflets, 30 and 90 min after spraying with 15 mg L^−1^ ZnO@OAm NRs or distilled water (control), using an Imaging PAM Fluorometer M-Series MINI-Version (Heinz Walz GmbH, Effeltrich, Germany), as described in detail previously [125]. Fluorescence was excited by a blue LED with a short-pass filter set to λ < 510 nm. Ten circular areas of interest (AOIs) were selected in each leaf to have the characteristic whole-leaf measurement. The actinic light (AL) used for the measurements was 600 μmol photons m^−2^ s^−1^, corresponding to the growth light (GL) of tomato plants, or 1000 μmol photons m^−2^ s^−1^, corresponding to high light (HL) intensity. The chlorophyll fluorescence parameters described in Appendix A were estimated using Win V2.41a software (Heinz Walz GmbH, Effeltrich, Germany).

### 4.7. Evaluation of Hydrogen Peroxide Generation

Hydrogen peroxide (H_2_O_2_) generation in tomato leaflets was evaluated 90 min after tomato plants were sprayed with 15 mg L^−1^ ZnO@OAm NRs or distilled water (control), as explained previously [44]. Leaves were incubated with 25 μM 2′, 7′-dichlorofluorescein diacetate (DCF-DA, Sigma Aldrich, Chemie GmbH, Schnelldorf, Germany) for 30 min in the dark and observed with a Zeiss AxioImager Z2 epi-fluorescence microscope equipped with an AxioCam MRc5 digital camera [105].

### 4.8. Statistical Analysis

Significant differences were determined using an ANOVA, followed by Tukey’s post hoc tests for each parameter. Before the tests, we evaluated the assumptions for raw data normality and we used the variance homogeneity test to verify the parametric distribution of the data. Values were considered significantly different at *p* < 0.05.

## 5. Conclusions

Foliar spray with 15 mg L^−1^ ZnO@OAm NRs decreased the efficiency of the oxygen-evolving complex (OEC), with a concomitant increase in ROS generation, a decrease in the maximum efficiency of PSII photochemistry (F*v*/F*m*), a decrease in the electron transport rate (ETR), and a decrease in the effective quantum yield of PSII photochemistry (Φ*_PSII_*), indicating reduced PSII efficiency. We conclude that rod-shaped ZnO@OAm NRs reduced PSII photochemistry, in contrast to the irregularly shaped ZnO@OAm NPs, which enhanced PSII efficiency [13]. In the transport of NPs into plant mesophyll tissues, it is essential to recognize that the process does not depend solely on the particle size but also on its shape and electrostatic characteristics. Thus, the morphology of NPs, including their shape and organic coating, determines factors which are important for their mechanism of action and their impact on crop yield.

## Figures and Tables

**Figure 1 plants-12-03502-f001:**
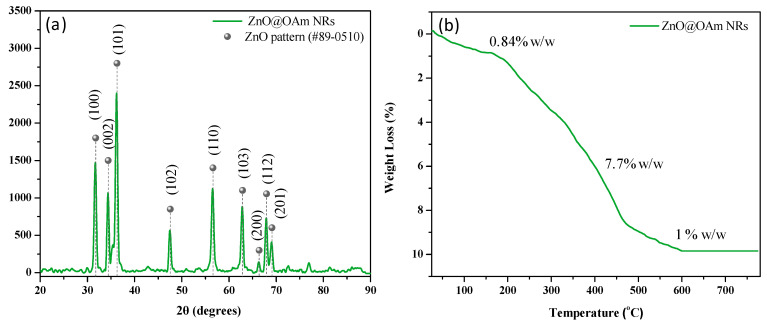
X-ray diffraction (XRD) pattern (**a**) and thermogravimetric analysis (TGA) curve (**b**) of the ZnO@OAm NRs.

**Figure 2 plants-12-03502-f002:**
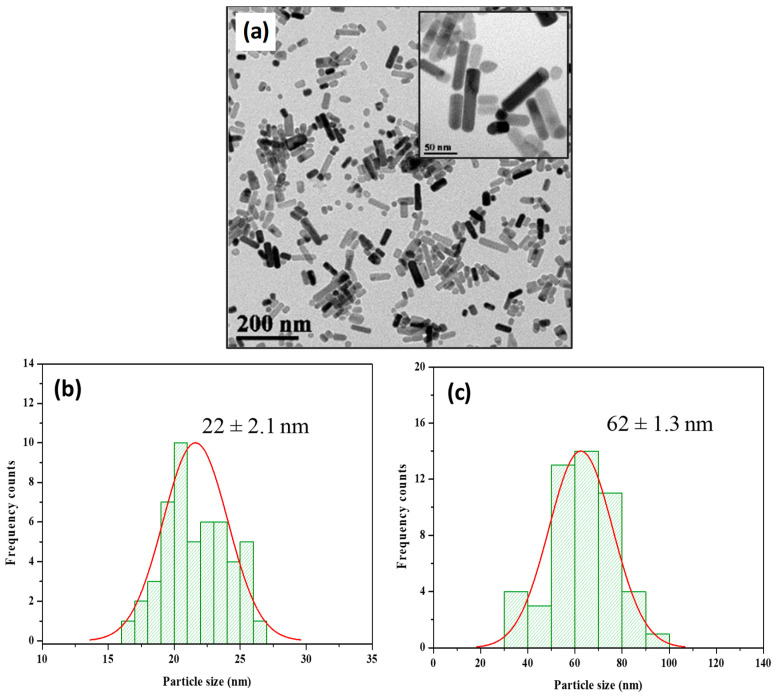
Transmission electron microscopy (TEM) image of ZnO@OAm NRs (**a**), histogram of average width and Gaussian fit (22 ± 2.1 nm) (**b**), and average length histogram and Gaussian fit (62 ± 1.3 nm) (**c**). The average size of particles is expressed as mean size (±SD).

**Figure 3 plants-12-03502-f003:**
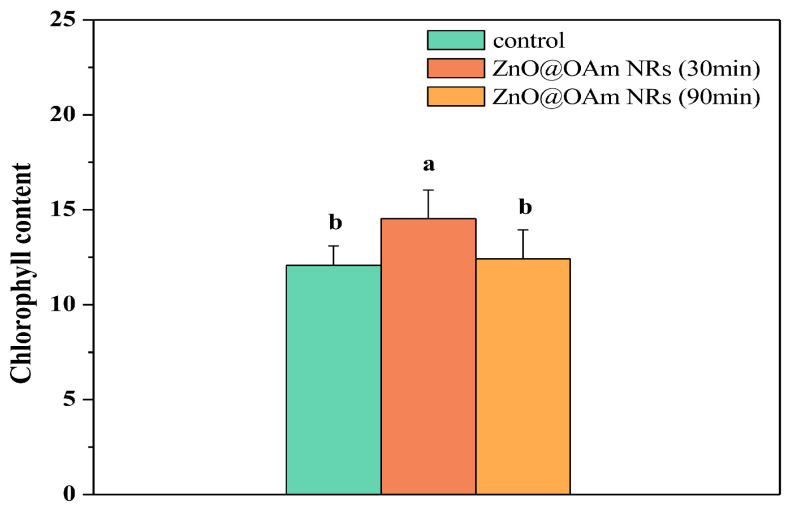
The chlorophyll contents of control (sprayed with distilled water) tomato leaflets and tomato leaflets 30 and 90 min after spraying with 15 mg L^−1^ ZnO@OAm NRs. Different lowercase letters denote statistical difference (*p* < 0.05). Error bars in columns are SDs.

**Figure 4 plants-12-03502-f004:**
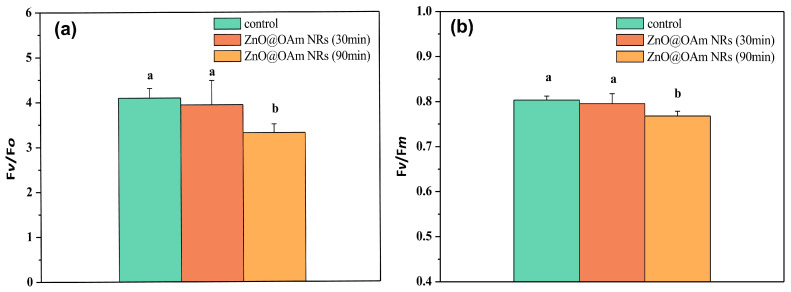
The efficiency of the oxygen-evolving complex (OEC) (F*v*/F*o*) (**a**) and the maximum efficiency of PSII photochemistry (F*v*/F*m*) (**b**) of control (sprayed with distilled water) tomato leaflets and tomato leaflets 30 and 90 min after spraying with 15 mg L^−1^ ZnO@OAm NRs. Different lowercase letters denote statistical difference (*p* < 0.05). Error bars in columns are SDs.

**Figure 5 plants-12-03502-f005:**
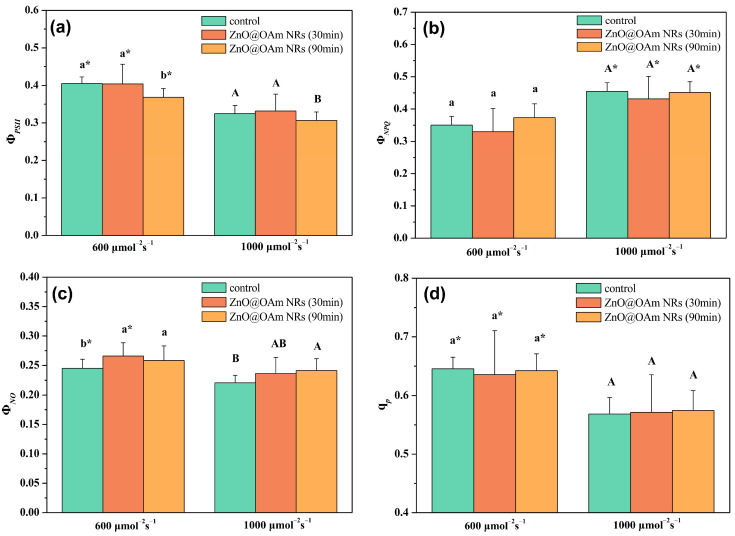
The effective quantum yield of PSII photochemistry (Φ*_PSII_*) (**a**), the quantum yield of regulated non-photochemical energy loss in PSII (Φ*_NPQ_*) (**b**), the non-regulated energy loss in PSII (Φ*_NO_*) (**c**), and the fraction of open PSII reaction centers (**d**) of control (sprayed with distilled water) tomato leaflets and tomato leaflets 30 and 90 min after spraying with 15 mg L^−1^ ZnO@OAm NRs. Statistical differences (*p* < 0.05) at the growth light (600 μmol photons m^−2^ s^−1^) intensity are indicated by different lowercase letters, while statistical differences at the high light (1000 μmol photons m^−2^ s^−1^) intensity are indicated by capital letters. The asterisks (*) represent significant differences (*p* < 0.05) between the growth light intensity and the high light intensity for the same treatment. Error bars in columns are SDs.

**Figure 6 plants-12-03502-f006:**
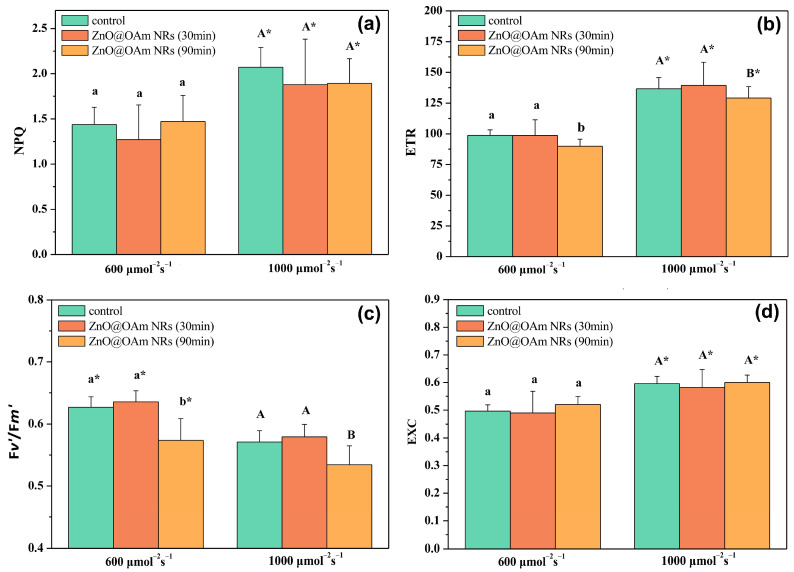
The heat dissipation by non-photochemical quenching (NPQ) (**a**), the electron transport rate (ETR) (**b**), the efficiency of open PSII reaction centers (**c**), and the excess excitation energy at PSII (**d**) of control (sprayed with distilled water) tomato leaflets and tomato leaflets 30 and 90 min after spraying with 15 mg L^−1^ ZnO@OAm NRs. Statistical differences (*p* < 0.05) at the growth light (600 μmol photons m^−2^ s^−1^) intensity are indicated by different lowercase letters, while statistical differences at the high light (1000 μmol photons m^−2^ s^−1^) intensity are indicated by capital letters. The asterisks (*) represent significant differences (*p* < 0.05) between the growth light intensity and the high light intensity for the same treatment. Error bars in columns are SDs.

**Figure 7 plants-12-03502-f007:**
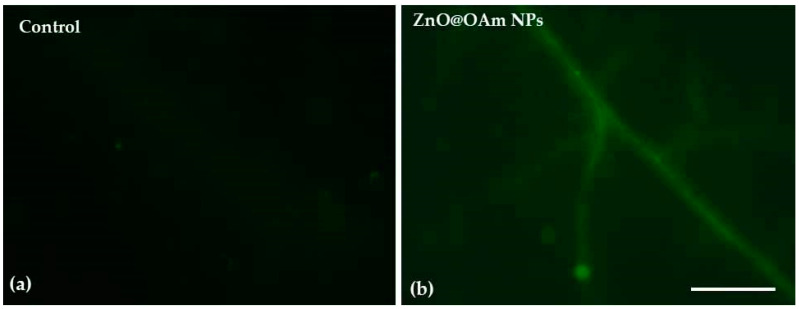
Imaging of H_2_O_2_ production on tomato leaflets 90 min after spraying with distilled water (control) (**a**) or 15 mg L^−1^ ZnO@OAm NRs (**b**). The light green color denotes the H_2_O_2_ generation. Scale bar: 500 μm.

## Data Availability

The data presented in this study are available in this article.

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
