# Peer review of "Modification of Tomato Photosystem II Photochemistry with Engineered Zinc Oxide Nanorods"

_plants, 2023, doi:10.3390/plants12193502_

Round 1

Reviewer 1 Report

The idea of improving tomato photosynthetic efficiency through engineering zinc oxide nanorods is very novel in this article. But this article needs to address several issues before it can be published.

1, The idea of improving tomato photosynthetic efficiency through engineering zinc oxide nanorods is very novel in this article. But this article needs to address several issues before it can be published.

2, The most direct evidence of effectiveness is the yield or biomass of crops. Why did the author not provide photos of different treatments or test the biomass of plants?

3, The latest literature on the impact of many nanomaterials on plant chlorophyll content and photosynthetic efficiency needs to be fully discussed and cited.

Author Response

1, The idea of improving tomato photosynthetic efficiency through engineering zinc oxide nanorods is very novel in this article. But this article needs to address several issues before it can be published.

We revised our manuscript taking into account your suggestions.

2, The most direct evidence of effectiveness is the yield or biomass of crops. Why did the author not provide photos of different treatments or test the biomass of plants?

The purpose of our research (L105-110) was to compare the newly engineered rod-shaped ZnO@OAm NRs with the previously examined irregularly shaped ZnO@OAm NPs [13] for their behavior on photosynthetic function and ROS generation. For this reason, we used the same plant material (tomato), the same exposure time (30 min and 90min), and the same concentration (15 mg L−1), as in our previous article [13]. Exposure time of 30 and 90min could not result in any biomass difference.

3, The latest literature on the impact of many nanomaterials on plant chlorophyll content and photosynthetic efficiency needs to be fully discussed and cited.

We added a paragraph in Discussion section (L 351-362), with recent literature, referring to the impact of ZnO NPs on plant chlorophyll content and photosynthetic efficiency.

Reviewer 2 Report

Review comment on “Modification of Tomato Photosystem II Photochemistry with engineered Zinc Oxide Nanorods”

General comment,

It is a good article to discuss the use of engineered irregularly shaped zinc oxide nanoparticles (ZnO NPs) coated with oleylamine (OAm), as photosynthetic bio-stimulants, for enhancing crop yield. In the current research, we tested the newly engineered rod-shaped ZnO nanorods (NRs), coated with oleylamine, (ZnO@OAm NRs), regarding their in vivo behavior on photosynthetic function and reactive oxygen species (ROS) generation, on tomato (Lycopersicon esculentum Mill.) plants. Well, I think it is good research for plants with the topic of NPs application in the agricutlrue. But I am not sure, only 90 minutes is enough to reflect the adventure of the NPs? Or we need a longer time to do the continuous detection. Please also check the following comments for the major revision.

Comments

1.     I am not sure where is your research hypothesis, a hypothesis is important for your study, please highlight it in the Introduction;

2.     I am not sure it is good to describe ZnO@OAm as a new material, I think it is a ZnO based material, and please describe and discuss the results with more properties of ZnO;

3.     How you treated the samples with your NPs? Please describe more in your M&M;

4.     Stomata is located more on the back of the leaf, and usually, it may need a longer time after the exposure for the NPs transfer, do you have some pre-experiment for the evidence that the NPs can touch the chloroplast after exposure;

5.     The reason for your light intension selection is needed. 

I think the language is fine, but please be series for some scientific writing skills. 

Author Response

General comment,

It is a good article to discuss the use of engineered irregularly shaped zinc oxide nanoparticles (ZnO NPs) coated with oleylamine (OAm), as photosynthetic bio-stimulants, for enhancing crop yield. In the current research, we tested the newly engineered rod-shaped ZnO nanorods (NRs), coated with oleylamine, (ZnO@OAm NRs), regarding their in vivo behavior on photosynthetic function and reactive oxygen species (ROS) generation, on tomato (Lycopersicon esculentum Mill.) plants.

Well, I think it is good research for plants with the topic of NPs application in the agriculture. But I am not sure, only 90 minutes is enough to reflect the adventure of the NPs? Or we need a longer time to do the continuous detection. Please also check the following comments for the major revision.

We revised our manuscript taking into account your suggestions.

The purpose of our research (L105-110) was to compare the newly engineered rod-shaped ZnO@OAm NRs with the previously examined irregularly shaped ZnO@OAm NPs [13] for their behavior on photosynthetic function and ROS generation. For this reason, we used the same exposure time (30 min and 90min).

We have in mind to evaluate in a future experiment the impact on photosynthesis of both rod-shaped ZnO@OAm NRs and irregularly shaped ZnO@OAm NPs after a longer exposure time (3 to 4 days).

Comments

  1. I am not sure where is your research hypothesis, a hypothesis is important for your study, please highlight it in the Introduction;

We rewrote the hypothesis in Introduction (L 105-110).

  1. I am not sure it is good to describe ZnO@OAm as a new material, I think it is a ZnO based material, and please describe and discuss the results with more properties of ZnO;

The size, shape, structure, and surface chemistry of engineered inorganic NPs, known as 4S, govern their efficiency [e.g., Nanotoxicology 2016, 10 (3), 257–278. https://doi.org/10.3109/17435390.2015.1048326.]. In particular, size and shape contribute to the NP's ability for successful attachment and entrance inside the cell. Thus, as a new material a thorough characterization is undertaken and given by XRD, TEM, TGA analysis, FTIR, UV-Vis, DLS and z-potential measurements.

  1. How you treated the samples with your NPs? Please describe more in your M&M;

We added in M&M (L411-413) that each tomato plant was foliar sprayed once with a volume of 10 mL of 15 mg L−1 ZnO@OAm NRs.

  1. Stomata is located more on the back of the leaf, and usually, it may need a longer time after the exposure for the NPs transfer, do you have some pre-experiment for the evidence that the NPs can touch the chloroplast after exposure;

Lower epidermis has more stomata than the upper epidermis but according to Zhu et al. [112] Zn concentration in the tissues close to the lower epidermis was higher than that in the upper epidermis when wheat plants were sprayed with ZnO NPs. However, mesophyll cells near the upper epidermis (parenchyma cells) have more chloroplasts than mesophyll cells near the lower epidermis (spongy cells). The evidence that ZnO@OAm NRs were translocated to the chloroplasts is the observed impact on photosystem II activity.

  1. The reason for your light intension selection is needed.

The purpose of our research (L105-110) was to compare the newly engineered rod-shaped ZnO@OAm NRs with the previously examined irregularly shaped ZnO@OAm NPs [13] for their behavior on photosynthetic function and ROS generation. For this reason, we used the same plant material (tomato), the same exposure time (30 min and 90min), the same concentration (15 mg L−1), and the same light intensities as in our previous article [13].

The selection of the light intensities is explained also in M & M (L 427-429). We used for the measurements the light intensity of 600 μmol photons m−2 s−1 that corresponded to the growth light (GL) of tomato plants, and that of 1000 μmol photons m−2 s−1 that corresponded to a high light (HL) intensity.

Round 2

Reviewer 1 Report

Authors have addressed all my comments.

Reviewer 2 Report

I think the revision is good. I think it can be accepted by current revision.